



# Water hyacinths as riverine plastic pollution carriers

Tim H.M. van Emmerik[1,*], Tim W. Janssen[1,*], Tianlong Jia[2], Thanh-Khiet L. Bui[3], Riccardo Taormina[2], Hong-Q. Nguyen[3], and Louise J. Schreyers[1]

[1]Hydrology and Environmental Hydraulics Group, Wageningen University, Wageningen, the Netherlands
[2]Department of Water Management, Delft University of Technology, Delft, the Netherlands
[3]Institute for Circular Economy Development, Vietnam National University-Ho Chi Minh City, Vietnam
[*]These author contributed equally to this work

**Correspondence:** Tim H.M. van Emmerik (tim.vanemmerik@wur.nl)

**Abstract.** Plastic pollution is an emerging entity threatening freshwater and marine ecosystems. Rivers play an important role in the transport and retention of plastic from land to sea. Tropical rivers are among the most polluted globally, and are assumed to emit substantial amounts of plastic into the marine environment. Concurrently, tropical rivers are invaded by water hyacinths, a free-floating vegetation species native to the Amazon. Water hyacinths grow rapidly, forming dense mats

of plants and other material including plastic pollution. With only limited anecdotal and scientific evidence of plastic-water hyacinth trapping, its full spatial extent along river systems remain unknown. Here, we demonstrate the consistent role of water hyacinths as carriers of plastic pollution along a river. Over 69k plastic items and 57k water hyacinth patches were identified along the 42 km most downstream section of the Saigon river, Vietnam. More than 73% of all floating plastics were carried by water hyacinths, ranging between 58-82% per specific location. The highest trapping ratio was found at the most upstream

locations. Although water hyacinths only covered 1.3% of the total river surface, the plastic concentration in water hyacinths was 197 times higher than in open water. Most downstream, the lowest water hyacinth coverage (0.2%) corresponded to the largest difference between surface plastic concentration in water hyacinths and at the open water (factor 781). Previous work demonstrated the effective trapping of plastic pollution by water hyacinths at individual sites. Here, we show that plastic-water hyacinth aggregates consistently occur at the river scale. We quantified plastic and water hyacinths at five locations along the

Saigon river, Vietnam, using drones and fixed cameras, in combination with a custom-trained YOLOv8 deep learning model. With our paper, we support the theory that water hyacinths effectively concentrate and carry plastic pollution along rivers. Further work on plastic-water hyacinth interactions is needed to better understand the transport, fate and impact of plastic in the world's most polluted rivers. Our results also support the idea of plastic monitoring from space using well-detectable floating vegetation as a proxy. Finally, our work suggests that current removal practices of water hyacinths may be optimized

to also recover plastic pollution from rivers.





# 1 Introduction

Rivers in the world's tropical regions are main contributors to global river plastic pollution and emissions into the ocean (Meijer et al., 2021; Roebroek et al., 2021). Despite the large uncertainties in global model based estimates, rivers in South(east) Asia,

Central and South America, and Sub-Sahara Africa consistently rank among the most polluted (González-Fernández et al., 2023). In light of global efforts to end plastic pollution, such hotspots are important focus regions for prevention and reduction measures (Simon et al., 2021; Tasseron et al., 2024). Tropical rivers are also excellent habitats for the invasive water hyacinths. Originating from the Amazon, water hyacinths have spread rapidly around the world (Penfound and Earle, 1948; Madikizela, 2021). Water hyacinths are free-floating aquatic plants with freely hanging roots, making them very mobile in response to

water flow and wind forcing. Their rapid growth is mainly driven by nutrient load, forming large interwoven mats (May et al., 2022). Due to their negative impacts, including clogging water infrastructure, deteriorating water quality, and blocking waterways, water hyacinths are considered an invasive pest and actively removed from water bodies (Jirawattanasomkul et al., 2021; García-de Lomas et al., 2022). Several anecdotal and scientific reports have described the trapping of plastic within water hyacinths plants and mats (van Emmerik et al., 2019; Schreyers et al., 2021a; Pritasari Arumdati, 2021; Pajai, 2022; van

Emmerik et al., 2022). Disentangling the interactions between plastic and water hyacinths is necessary to better understand transport and fate of plastic pollution along river systems.

Interactions between water hyacinths and plastic pollution have been reported for rivers globally, including in Thailand, Indonesia, the Dominican Republic, and Vietnam. Most work on describing plastic-water hyacinth interactions has been done on the Saigon river, Vietnam. In an effort to explain the seasonality in floating plastic transport in the Saigon, (van Emmerik et al.,

2019) found the strongest correlation between plastic transport and organic material concentration rather than river discharge or tidal dynamics. The organic material mainly consisted of water hyacinths and parts thereof, leading to the hypothesis that water hyacinths play an important role in river plastic transport dynamics. To test this hypothesis, (Schreyers et al., 2021b) developed several measurement methods to quantify the accumulation of plastic water hyacinths, ranging from physical sampling to visual counting and image-based techniques. Two field-based studies showed that for two different cross-sections,

between 54-78% of the total floating plastic items were transported within water hyacinth (Schreyers et al., 2021a, 2024a). These results fueled the hypothesis that the plastic trapping mechanisms are consistent along the Saigon river. A follow-up study released GPS trackers to mimic floating plastic items, and found that over 80% of retrieved items got stuck in water hyacinths along the 40 km river reach studied (Lotcheris et al., 2024). Although this work showed the spatial extent of the plastic-water hyacinth interactions, no additional data on hyacinth coverage and plastic surface concentrations were collected.

Although there is increased evidence of the consistent interactions between plastic pollution and water hyacinths, the spatial extent of the plastic trapping effect of water hyacinths remains unclear. In this paper, we therefore present a first assessment of plastic-water hyacinth interactions along the downstream 42 km reach of the Saigon river.

Building on previous assessments, we further explore the role of water hyacinths as plastic carriers at the river scale. To this aim, data was collected from multiple sensors, including fixed cameras and Uncrewed Aerial Vehicles (UAV). A total of

55 14,925 images were collected to detect plastic and water hyacinths, making it one of the largest river plastic image datasets to



date. Only for the marine environment larger image datasets have been used, covering much larger areas (Jia et al., 2023). We used a custom-trained You Only Look Once v8 (YOLOv8) deep learning model to identify plastic items and water hyacinths, and quantify water hyacinth patch size. We quantified water hyacinth coverage, plastic surface concentrations within water hyacinths and at the open water surface, and the size distribution of water hyacinths at five locations along the Saigon river. Analyzing these variables in parallel also allowed to develop a more holistic understanding of the interactions between plastic and water hyacinths along rivers.

The goal of this paper is to explore the variability of plastic-water hyacinth trapping along the Saigon river. Our findings emphasize that plastic-water hyacinth interactions are an integral component of the plastic transport and retention dynamics in tropical rivers. We argue that this is relevant for three reasons. First, to better understand the fate, transport and impact of plastic in (tropical) rivers, the interactions between water hyacinths (or other species) and plastic is crucial (van Emmerik and Schwarz, 2020). Second, the consistent co-occurrence of plastic pollution and water hyacinths can be used to develop plastic detection methods from space using water hyacinths as a proxy (Schreyers et al., 2022). Finally, the ability to effectively concentrate plastic paves the way to explore the use of water hyacinths in developing plastic pollution reduction strategies. We encourage future work to focus on assessing the interactions between water hyacinths and plastic in other river systems, and between different floating vegetation species in rivers globally.

## 2  Methods

We used image-based techniques to quantify water hyacinth coverage, plastic concentration, and plastic-hyacinth aggregations along the Saigon river, Vietnam, during the dry season of 2023 (Section 2.1). Images were collected using bridge mounted cameras and UAVs at five locations along a 42 km-long river section that flows through Ho Chi Minh City (Section 2.2). We used the YOLOv8 (Jocher et al., 2023) deep learning algorithm to detect water hyacinth patches and plastic items in each image, trained on a subset with annotated images from the collected database (Section 2.3). Note that we used "plastic" for all anthropogenic macrolitter items larger than 2.5 cm, as the vast majority ( 85%) of the litter items and mass are plastic in the Saigon (van Emmerik et al., 2019). We assumed a minimum detectable item size of around 2.5 cm, based on the height and specifications of the UAV and cameras (Section 2.2). We used the image-based data to assess the role of water hyacinths as carriers of plastic along the Saigon.

### 2.1  Study area

The Saigon river is approximately 225 km long and drains an area of around 4,800 km$^2$. It traverses Ho Chi Minh City (over 10 million inhabitants), and joins the Dong Nai river before flowing into the ocean. The climate is characterized by a wet (May to November) and dry season (December to April). The river flow is influenced by the upstream Dau Tieng reservoir management, freshwater inflow, and tidal dynamics causing flow reversal. The discharge varies between -1500 and 2000 m$^3$/s, and the tidal amplitude reaches 4 m during spring tide (Camenen et al., 2021; Rodrigues do Amaral et al., 2024). The tidal



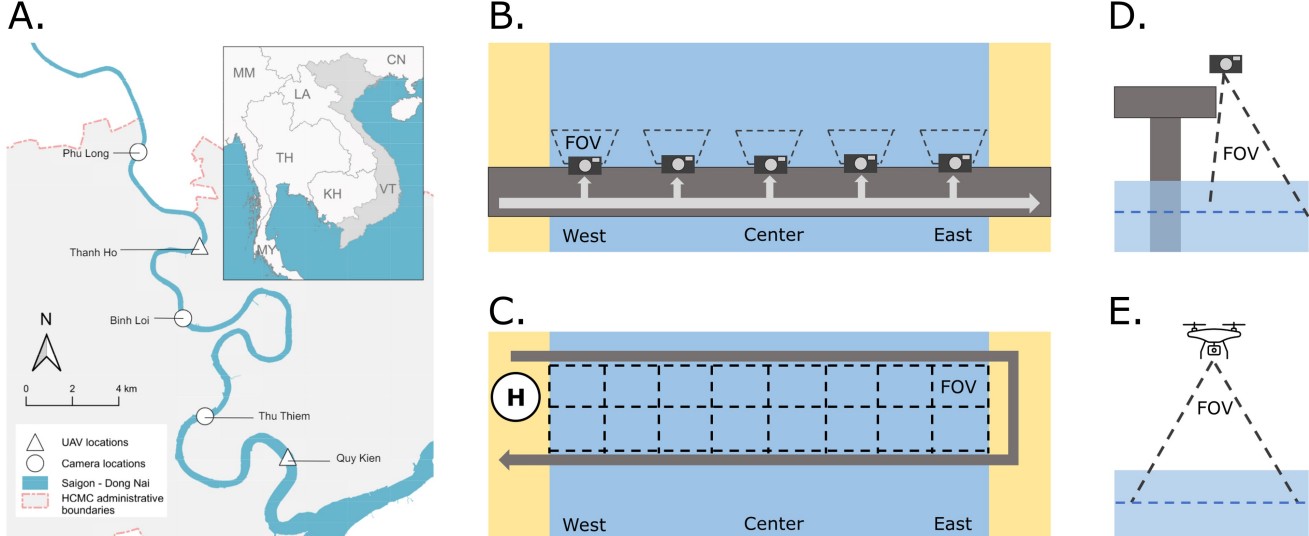

**Figure 1.** A. Overview of the study area, including the five measurement locations ranging from Phu Long to Quy Kien, which are 46.9 to 4.9 km upstream of the Saigon-Don Nai confluence, respectively. The triangles indicate the camera locations, and the crosses the UAV locations (base map: ESRI). B. The fixed camera setup, including the five locations and Field of View (FOV). C. The UAV setup, including flight path, the FOV, and landing zone (H). D. The side-view of the camera setup, including FOV. E. The side-view of the UAV setup, including FOV.

regime is asymmetrical, mixed semi-diurnal, resulting in flow reversal twice per day (Schreyers et al., 2024b). The Saigon is affected by invasive floating water hyacinths, typically peaking between December and May (Janssens et al., 2022).

## 2.2 Field data collection

The measurements were done during the dry season, between 6 February and 1 April 2023 at five locations along the Saigon river (Table 1). The most upstream (Phu Long) and downstream (Quy Kien) locations were 41.9 km and 5.5 km upstream of the Dong Nai confluence, respectively. We used bridge mounted fixed cameras and UAVs to estimate water hyacinth coverage and patch size, plastic concentrations at the water surface and plastic concentrations within water hyacinths. Three locations were measured using bridge mounted cameras (Phu Long, Binh Loi, Thu Thiem), and two were measured using UAVs (Thanh

Ho, Quy Kien) as no suitable bridges were available. We defined four measurement periods with a duration of two weeks. The camera measurements were done in all four measurement periods, and the UAV measurements in period 1, 3 and 4.

### 2.2.1 Bridge mounted cameras

We used a GoPro Hero 11 (GoPro, Inc., San Mateo, USA) RGB camera to collect images at Phu Long, Binh Loi, and Thu Thiem (Tabl2 ). The camera was attached to an extendable 0.40 m arm, which was mounted pointed downwards to the railing

of the bridge. We used the linear 8:7 camera mode to mitigate the camera barrel distortion effect (spherizing of the image) (Lee



**Table 1.** Overview of the measurement locations, including distance to the Saigon-Dong Nai confluence, the measurement method, and the number of observations per round.

| | Location | | Distance to Dong Nai | River width | Method | Observations per round | | | |
|---|---|---|---|---|---|---|---|---|---|
| | | | | | | 1 | 2 | 3 | 4 |
| Name | Lat | Lon | [km] | [m] | | 6-19 Feb | 20 Feb-5 Mar | 6-19 Mar | 20 Mar - 1 Apr |
| Phu Long | 10.890382 | 106.69196 | 41.9 | 185 | Camera | 6 | 6 | 6 | 6 |
| Thanh Ho | 10.853241 | 106.71632 | 35.9 | 210 | UAV | 6 | 0 | 21 | 12 |
| Binh Loi | 10.825805 | 106.70908 | 30.4 | 225 | Camera | 6 | 4 | 7 | 6 |
| Thu Thiem | 10.785911 | 106.71836 | 14.8 | 305 | Camera | 4 | 2 | 8 | 6 |
| Quy Kien | 10.770056 | 106.75085 | 5.5 | 480 | UAV | 1 | 0 | 10 | 10 |

et al., 2019). To reduce the obstruction of the bridge structure on the Field-of-View (FOV), the camera angle deviated slightly from nadir (standard at $10°$, $30°$ for Thu Thiem during period 1). We selected five measurement points across each bridge. The two outer points (East, West) were chosen such that the camera FOV bordered the riverbank. The middle point (Center) was selected in the center of the river. The remaining two points (Mid-East, Mid-West) were chosen in between the Center and East/West points. We used a single camera for all measurements, which was only mounted to the bridge for the duration of a single measurement. On each measurement day, up to eight measurements rounds were done, distributed between morning and afternoon as equally as possible. For each round, we moved from the West point towards the East. At each observation point we took 31 images with an interval of 10 seconds. A single round took between 45 to 60 minutes. The distance to the water surface varied between 7.4-18.6 m, depending on the location and the tidal phase. In total, 11,297 camera images were collected.

The Ground Sampling Distance ($d_g$) [cm/pixel] (Geraeds et al., 2019) of every image was calculated using the distance from the camera to the water surface $H$ [m], the sensor width $w_s$ [mm], the image size $S_i$ [pixels] and the camera focal length $L_f$ [mm]:

$$d_g = \frac{H \cdot w_s}{S_i \cdot L_f} \tag{1}$$

Table 2 presents an overview of all minimum and maximum $d_g$ for each measurement location.

### 2.2.2 Uncrewed Aerial Vehicles

We used a DJI Phantom 4 Pro (SZ DJI Technology Co., Ltd., Shenzhen, China) to take RGB images at Thanh Ho and Quy Kien (Table 2). For a single flight, the UAV crossed the entire river width back and forth following a U-shape flight path (Geraeds et al., 2019). Images were taken along the entire river width in both flight directions. The images on the back and forth crossing had 30% overlap, and adjacent images taken at each crossing had 20% overlap. Images were taken during the flight/while



**Table 2.** Details of the camera height, resolution ($d_g$), total and annotated images, and annotated items per location.

| Location | Camera height [m] | | $d_g$ [cm/pixel] | | FOV [m2] | | Total images | Annotated images | Annotated items |
|---|---|---|---|---|---|---|---|---|---|
| Name | min | max | min | max | min | max | | | |
| Phu Long | 9.5 | 12.9 | 0.35 | 0.48 | 569 | 1049 | 3613 | 65 | 2024 |
| Thanh Ho | 11 | 12.0 | 0.3 | 0.46 | 181 | 266.5 | 1851 | 24 | 775 |
| Binh Loi | 7.4 | 9.9 | 0.28 | 0.37 | 345 | 618 | 3691 | 65 | 1559 |
| Thu Thiem | 11 | 18.6 | 0.41 | 0.69 | 763 | 4802 | 3941 | 93 | 4741 |
| Quy Kien | 11 | 14.0 | 0.3 | 0.46 | 181 | 294 | 2399 | 11 | 266 |

hovering, and the velocity was 2 m/s. Images were taken at nadir with a distance of 11-14 m to the water surface, depending on the tidal phase. The deployment and landing locations were kept constant for all flights. We conducted one to nine flights per measurement day, depending on weather conditions and battery status. The mean number of images per flight varied per location (Thanh Ho: 51; Quy Kien: 130), mainly due to the variation in river width (Thanh Ho: 210 m; Quy Kien: 480 m). In total, 4,306 UAV images were collected.

### 2.2.3 Image processing

The camera images were corrected for the keystone effect, i.e., the distortion of the dimensions caused by not taking the images at nadir. The keystone effect increasingly reduced the $d_g$ towards the far edges of the field-of-view (upper left and right). Along the horizontal plane the distortion is constant. We corrected for the keystone effect by calculating a correction factors for the off-nadir angles (10°, and 30°). For the image annotation and deep learning model (Section 2.3) purposes, each image was divided into 8x8 tiles. We therefore calculated the correction factor for each of the eight vertical rows. First, we calculated the true coordinates of the four image corners based on the aspect ratio (8:7), the vertical (100°) and horizontal (133°) lens angle, and the deployment angle, using trigonometry. Second, the images were stretched using open source photo editing program GNU Image Manipulation Program (GIMP). Finally, we calculated the corrected number of pixels and surface area per tile from the stretched images. The UAV images were taken at nadir and did not need keystone correction.

## 2.3 Image-based plastic and hyacinth detection

### 2.3.1 Deep learning model architecture description

We used the state-of-the-art YOLOv8 deep learning model architecture to detect water hyacinths and plastic items in camera and UAV images. The YOLO series networks, especially the new YOLOv8 network released in 2023, are popular architectures for object detection thanks to their high accuracy and fast inference speed (Terven and Cordova-Esparza, 2023).

YOLOv8 is a fully convolutional neural networks that consists of two main modules: (1) the backbone, and (2) the head. The backbone extracts relevant features at various resolution levels from the input images. These features are then processed



by the head module which detects object locations with bounding boxes and classifies each object independently. In the output layer of the architecture, YOLOv8 uses a softmax activation function to output the probabilities of a detected object belonging to every possible class in the training dataset. The most significant advancement in YOLOv8 compared to its predecessors is its anchor-free design. Instead of predicting offsets from predefined anchor boxes, YOLOv8 directly predicts the center of objects before constructing the related bounding boxes. This approach leads to better efficiency and improves generalization by allowing the model to adapt more effectively to a wide range of object sizes (Terven and Cordova-Esparza, 2023).

### 2.3.2 Image subsets for model development

We annotated objects with bounding boxes using the *Roboflow* platform (Dwyer et al., 2022). We labeled 272 images from the total of 15,495 images (72.9% camera, 27.4% UAV), with 9,352 object annotations. We identified three object categories: (1) water hyacinths, (2) free-floating plastic item, and (3) trapped plastic item within water hyacinths. The water hyacinth class objects include water hyacinths with and without entrapped plastic items. Free-floating plastic items are defined as plastic items freely floating at the water surface, and not trapped or in contact with water hyacinths. The trapped plastic items are defined as items clearly inside or in contact with water hyacinth. Of the total 9,352 annotated objects, 3,017 are water hyacinths, 2,036 are free-floating plastic items, and 4,299 are trapped plastic items, as reported in Table 3.

**Table 3.** The subsets used for model development. The model "resize" was used for water hyacinths, and "tiles" was used for plastic items.

| Model | Subsets | No. images or image tiles | No. annotated items for each class | | | No. total annotated items |
|---|---|---|---|---|---|---|
| | | | Water hyacinth | Free-floating plastic | Trapped plastic | |
| Model$_{resize}$ | Train$_{resize}$ | 218 | 2,040 | 1,437 | 3,453 | 6,930 |
| | Validation$_{resize}$ | 27 | 451 | 219 | 392 | 1,062 |
| | Test$_{resize}$ | 27 | 526 | 380 | 454 | 1,360 |
| | Total | 272 | 3,017 | 2,036 | 4,299 | 9,352 |
| Model$_{tiles}$ | Train$_{tiles}$ | 11,408 | 3,094 | 1,512 | 3,683 | 8,289 |
| | Validation$_{tiles}$ | 1,416 | 687 | 237 | 413 | 1,337 |
| | Test$_{tiles}$ | 1,440 | 732 | 398 | 436 | 1,566 |
| | Total | 14,264 | 4,513 | 2,147 | 4,532 | 11,192 |

### 2.3.3 Model development

Due to the input image size limitation of 640x640 pixels for YOLOv8, we used two common image processing methods: (1) resizing the original images into 640x640 pixels, and (2) cutting images into tiles of 640x640 pixels before feeding them into





the model (Jia et al., 2023). Compared with image resizing, tiling retains higher resolutions, potentially enhancing the model's performance in detecting plastic items, especially of low dimensions (Wolf et al., 2020; Kylili et al., 2020). However, cutting images into small tiles results in water hyacinths being being divided over several tiles, since they are typically larger than the tile dimensions. This can significantly impact the quantification of the number of water hyacinth patches.

Considering the benefits of each processing method, we developed two models in this study: (1) Model$_{\text{resize}}$ and (2) Model$_{\text{tiles}}$,

using one of the two processing methods, respectively. First, we resized the 272 labeled input images into 640x640 pixels, and divided them into Train$_{\text{resize}}$, Validation$_{\text{resize}}$, and Test$_{\text{resize}}$ subsets, following the 80/10/10 split detailed in Table 3. These three subsets are used for the Model$_{\text{resize}}$ training, validation and testing, respectively. To develop Model$_{\text{tiles}}$, we sliced the 272 images into tiles of 640×640 pixels. Each camera image was cut into 56 tiles, and each UAV image was cut into 32 tiles. This yielded 14,264 image tiles with annotated object items of interest (see Table 3). We divided these tiles into Train$_{\text{tiles}}$, Validation$_{\text{tiles}}$, and

Test$_{\text{tiles}}$ subsets, following the 80/10/10 split. These three subsets are used for the Model$_{\text{tiles}}$ development. The implementation of YOLOv8 is shown in Appendix A.

### 2.3.4   Evaluation metrics

We evaluated the models performance using three commonly employed metrics: (1) (m)AP50, (2) precision $P$, and (3) recall $R$ (Jia et al., 2023; Padilla et al., 2020). AP50 is the mean Average Precision (AP) of one class with an Intersection over Union

(IoU) threshold of 50%. The IoU quantifies the prediction accuracy by dividing the overlapping area of the predicted and ground-truth bounding boxes $A_I$, by their total covered area $A_U$. mAP50 is the average of AP50 across all classes. Precision is an accuracy measure of positive detections, denoted by the portion of correctly detected positive cases (i.e., true positives) out of the total number of cases detected as positive. Recall represents the ratio of correctly detected positive cases to the total number of ground-truth cases. We used the precision and recall computed with an IoU threshold of 0.5 to evaluate model

performance. The equations of these performance metrics can be found in Appendix B.

### 2.3.5   Model performance evaluation

We selected the developed Model$_{\text{resize}}$ to detect and quantify the number of hyacinths, and the Model$_{\text{tiles}}$ to accurately detect and quantify the number of plastic items on the entire dataset. Appendix C shows precision, recall, and (m)AP50 of (1) the Model$_{\text{resize}}$ on the Test$_{\text{resize}}$ subset, and (2) the Model$_{\text{tiles}}$ on the Test$_{\text{tiles}}$ subset. While the Model$_{\text{resize}}$ obtains high precision for

all classes ($P$ = 0.72-0.89) and high recall and AP50 for water hyacinth ($R$ = 0.54, and AP50 = 70%), it achieves significantly low recall for free-floating plastic ($R$ = 0.05) and trapped plastic ($R$ = 0.06). The results show that the Model$_{\text{resize}}$ performs well in detecting water hyacinth, but does not identify a large number of plastic items. This result is also shown in Fig C1 in Appendix C. Compared to the Model$_{\text{resize}}$, the Model$_{\text{tiles}}$ achieves relatively higher performance in detecting free-floating plastic ($P$ = 0.72, $R$ = 0.25, and AP50 = 48%), and trapped plastic ($P$ = 0.69, $R$ = 0.39, and AP50 = 52%). However, the Model$_{\text{tiles}}$

may count many hyacinth patches in image tiles as individual hyacinths, leading to an overestimation of their abundance.



### 2.3.6 Model output and metrics of interest

Two models were run on the entire dataset. Note that this is not common practice in machine learning, but our aim was to maximize data availability for the phenomenological analyses. The model runs yielded the (1) number of objects per class per image, and (2) coordinates and sizes of the object bounding boxes in pixels. We converted bounding box sizes from pixels to actual length using the $d_g$ (section 2.2.1) of each image [cm/pixel]. For each bounding box the surface area was reported in squared meters [m$^2$]. We applied the keystone correction to the camera images (section 2.2.3). For the water hyacinths we fitted ellipses inside each hyacinth bounding box using an ellipse factor of 0.79, to account for the predominant shape of water hyacinths (Schreyers et al., 2024a). We used the size of the retrieved bounding boxes as size for the free-floating and trapped plastic items in the rest of this study. For each image, we calculated the trapping ratio $r_{ent}$ [-], the water hyacinth river surface coverage $f_{wh}$ [-], the total river plastic surface concentration $C_r$ [#/km$^2$], the water hyacinth plastic surface concentration $C_{wh}$ [#/km$^2$], and the open water plastic surface concentration $C_o$ [#/km$^2$] (see Appendix D).

### 2.3.7 Statistical analysis

We used the Anderson-Darling test to test all relevant variables (section 2.3.6) for normality. We compared the mean and median values per site, measurement period, and flow direction using the Kruskall-Wallance (mean) and Wilcoxon (median) tests. We used $p = 0.05$ for the significance level.

## 3 Results and discussion

### 3.1 River plastic density varies along the river course

Water hyacinths consistently carried the majority (73%) of all floating plastics across locations and measurement periods. The site specific averages vary from 58% at Quy Kien (most downstream) to 82% at Phu Long and Tanh Ho (most upstream), with temporal variations per location up to 26% (Quy Kien) (Fig. 2). The highest trapping ratios were found in period 4, with the exception of Thu Thiem (period 3). The lowest trapping ratio was found for period 3 for the two upstream locations, and periods 1 and 2 for the three downstream locations. The highest trapping ratio (90%) was found in period 4 at Thanh Ho.

Our results support the hypothesis that water hyacinths act as important carriers of floating macroplastics in tropical rivers (van Emmerik et al., 2019). Previous work found trapping ratios between 54-78% based on observations taken between April 2020 and January 2022 (Schreyers et al., 2021a, 2024a). These studies focus only on Thu Thiem, which now showed the lowest mean trapping ratio. These findings therefore suggests that water hyacinths are not only relevant locally, but play an important role in plastic transport and retention dynamics at along a larger part of the river.

The overall mean plastic surface concentration is nearly 200 times larger within water hyacinths (2.7·10$^5$ #/km$^2$) than in the open water (1.4·10$^3$ #/km$^2$) . The water hyacinth plastic surface concentration $C_{wh}$ [#/km$^2$] is one (Binh Loi) to two (Quy Kien) orders of magnitude larger than the total river plastic surface concentration $C_r$ [#/km$^2$], and an additional order of magnitude larger than the open water plastic surface concentration $C_o$ [#/km$^2$] (Fig. 3). $C_{wh}$ shows an increasing trend in downstream





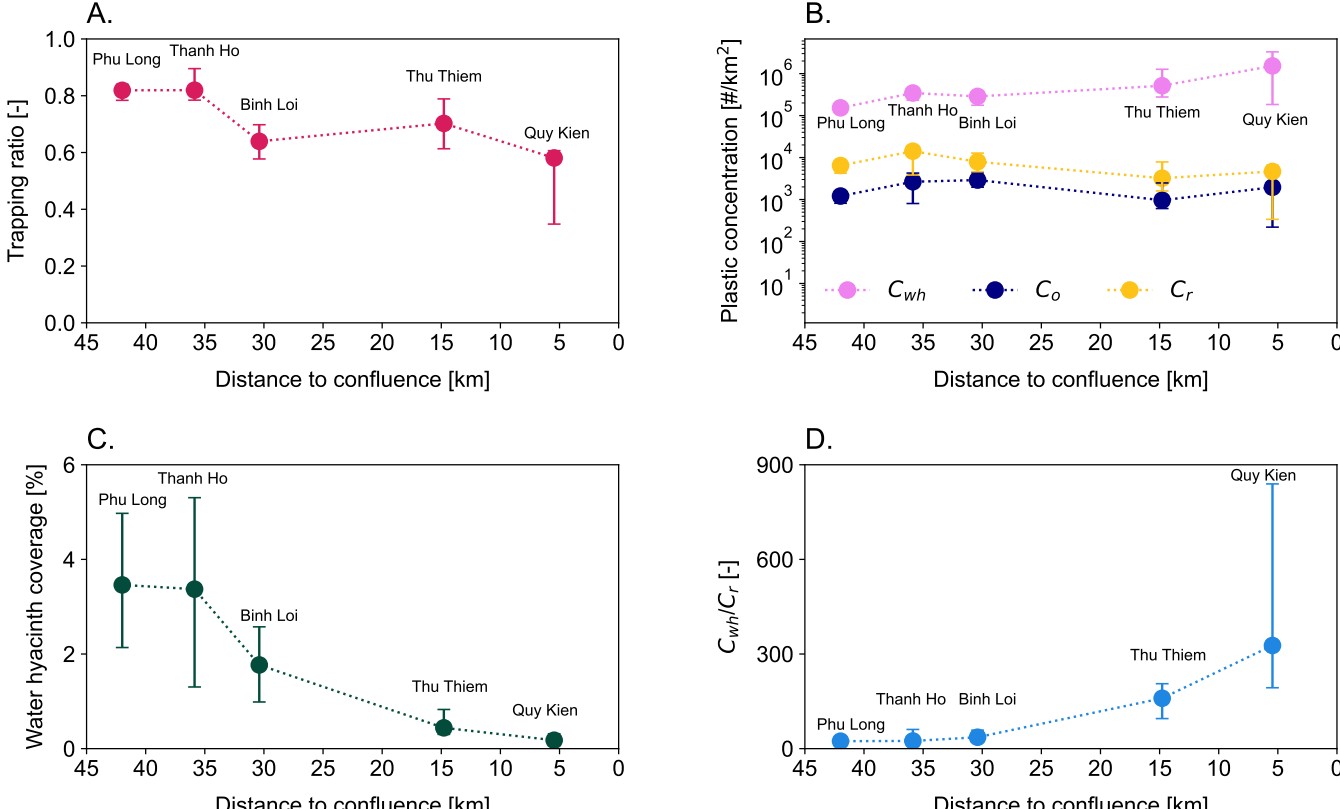

**Figure 2.** A. Plastic trapping ratio was on average 73%. The mean trapping ratio per location varied between 58 and 82%. Overall the highest trapping ratio were found at the most upstream locations, and the lowest at the most downstream location closest to the confluence. B. The surface plastic concentration [#/km$^2$] in water hyacinths increased towards the river mouth. In contrast, the total river plastic concentration [#/km$^2$] and open water surface plastic concentration [#/km$^2$] decreased towards the river mouth. C. The water hyacinth coverage of the river surface decreased towards the river mouth. The ratio between the surface plastic concentration in water hyacinths and at the total river plastic surface concentration.



direction, from $1.5\cdot10^5$ #/km$^2$ at Phu Long to $1.5\cdot10^6$ #/km$^2$ and Quy Kien. $C_r$ and $C_o$ decrease from the upstream locations (Phu Long to Binh Loi) towards the downstream locations (Thu Thiem and Quy Kien).

Schreyers et al. (2022) used a constant $C_{wh}$ of $2.1\cdot10^5$ #/km$^2$, which is in line with the values found at the three upstream

locations. Quy Kien however had a $C_{wh}$ of one order of magnitude larger. For specific river systems, a constant $C_{wh}$ may yield a reasonable first estimate, but may result in large uncertainties for individual locations. Understanding the spatial and temporal variation in $C_{wh}$ are therefore crucial for plastic detection methods that use water hyacinths as a proxy.

These results emphasize the important role of water hyacinths as carriers of floating macroplastics at the river scale. Along the entire studied river reach, the surface concentrations were 1 to 3 orders of magnitude higher in water hyacinths compared

to the total or open water surface concentrations. Water hyacinths seem to effectively entrap and concentrate floating plastic, leading to the much higher concentrations than when plastic items remain freely floating. The river width increases towards the confluence, which explains the decrease in $C_r$ and $C_o$. Interestingly, the $C_{wh}$ is the highest in the most downstream regions. This is explained by decrease in hyacinth coverage in combination with only a low decrease in trapping ratio. Downstream, a similar amount of plastic is trapped by a smaller water hyacinth area compared to upstream. As these are also the least

accessible areas for conventional plastic monitoring, this emphasizes the potential of using water hyacinths as a proxy for plastic pollution. Furthermore, it suggests that the water hyacinths that travel all the way down to the confluence have accumulated the most plastics. Removing water hyacinths in the downstream regions may therefore also lead to capturing the largest amount of plastics.

### 3.2    Hyacinth coverage decreases downstream

The water hyacinth coverage ranged between 3.5% (Phu Long, most upstream) and 0.2 % (Quy Kien, most downstream). At the three upstream locations the coverage varied between 1.0% and 5.3%, and at the two downstream locations between 0.1% and 0.8%. The overall mean coverage was 1.3%. There was no clear increase or decrease over time (Fig. 2C).

The mean surface area per water hyacinth was the highest at Tanh Ho (5.3 m$^2$), and the lowest at Quy Kien (1.6 m$^2$). The spatial trend of the mean water hyacinth area per patch follows the same overall decreasing trend towards the confluence. The

combined water hyacinth area statistics shows a strongly skewed distribution, with an overall mean of 3.3 m$^2$ and median of 0.5 m$^2$.

The one hundred largest water hyacinth patches varied from 140 to 401 m$^2$, 42-122 times the mean. Most of these extremely large water hyacinths were found at Phu Long (74) and Thanh Ho (5). The remaining ones were found at Binh Loi (5) and Thu Thiem (6). At the downstream location Quy Kien none were found.

The decreasing trend towards the confluence in water hyacinth coverage and mean surface area per water hyacinth patch is in line with the findings of Janssens et al. (2022). They used three years of Sentinel-2 data to look at the spatial and temporal trends of water hyacinth coverage. Here, the mean coverage was found between 3-24% for the river section similar to our study. The values between 5-24% were however only found for the sections upstream of Phu Long. The section close to the confluence, which comprises Thu Thiem and Quy Kien, had a long-term average coverage below 1.0%.



The low coverage in the downstream section may caused by both the increase in width, and the break-down of the water hyacinth patches (Petrell and Bagnall, 1991; Janssens et al., 2022). The river width increases from 185 m at Phu Long to 480 m at Quy Kien. With no changes in the water hyacinth area, this would already result in a factor 2.6 coverage decrease. However, our results show the decrease in mean water hyacinth surface area per patch from 5.3 to 1.6 $m^2$. These findings corroborate the suggestion that water hyacinths may disintegrate in the downstream sections due to higher flow velocity (variations).

### 3.3 Water hyacinths as consistent macroplastic carriers

Water hyacinths act as plastic carriers throughout the 42 km downstream section of the Saigon river (Fig. 2). Along this stretch, the river width increases from 185 m to 480 m, and traverses Ho Chi Minh City. The plastic trapping ratio was found relatively stable, with values between 63% and 82% along the river. Water hyacinth coverage, and water hyacinth mean area per patch decrease towards the river mouth. However, our results suggest that the role of water hyacinths as carrier of floating plastics is increasingly important towards the confluence. This is supported by the ratio in plastic surface concentration between the water hyacinths and in total at the river surface, increasing from 21-36 upstream to 306 near the confluence (Fig. 2c).

Although previous work on plastic-water hyacinth interactions has been done on the Saigon river, this is the first study that investigates these dynamics on the river scale at multiple locations. van Emmerik et al. (2019) hypothesized that water hyacinth abundance was driving total floating plastic transport at Thu Thiem. This was supported by Schreyers et al. (2024a) who used a one-year dataset collected at Thu Thiem to show that indeed up to 77% of the total plastic transport occurs through plastic items entrapped in water hyacinths. Lotcheris et al. (2024) used GPS-trackers released at Thu Thiem and Binh Loi to study the transport and retention dynamics. Over 80% of the retrieved items were trapped in water hyacinths. Based on our results we see that the high trapping ratio and much higher plastic concentrations in water hyacinths than in open water are spatially consistent. We therefore have increased confidence that water hyacinths play a key role in the river plastic transport and retention dynamics at larger spatial scales.

### 3.4 Uncertainties and limitations

Not all locations were covered equally during data collection. At Phu Long, Binh Loi and Thu Thiem we collected between 3,613 and 3,941 images. At Thanh Ho and Quy Kien only between 1,851 and 2,399 images. The total covered river surface varies even more, from 0.4 $km^2$ (Thanh Ho), to 8.3 $km^2$ (Thu Thiem). Note that the covered river surface is a function of both the number of images, and the height above the surface and the incidence angle of the camera. Thu Thiem has a much larger coverage due to the largest camera-to-river distance (up to 18.6 m) and a larger initial incidence angle (30° instead of 10°). The distance from the camera to the river surface varied due to water level variations, inaccuracies in the flight altitude, and variation in bridge height. The variation in image resolution ($d_g$ [cm/pixel]) is limited to a factor between 1.3 and 1.7 per specific locations. The variation between the minimum and maximum $d_g$ is a factor of 2.5. As a consequence, the minimum detectable plastic item size varied per image, introducing some uncertainty when comparing results from individual images. For the time and space averaged values, we do not expect a large impact on the found results and trends.



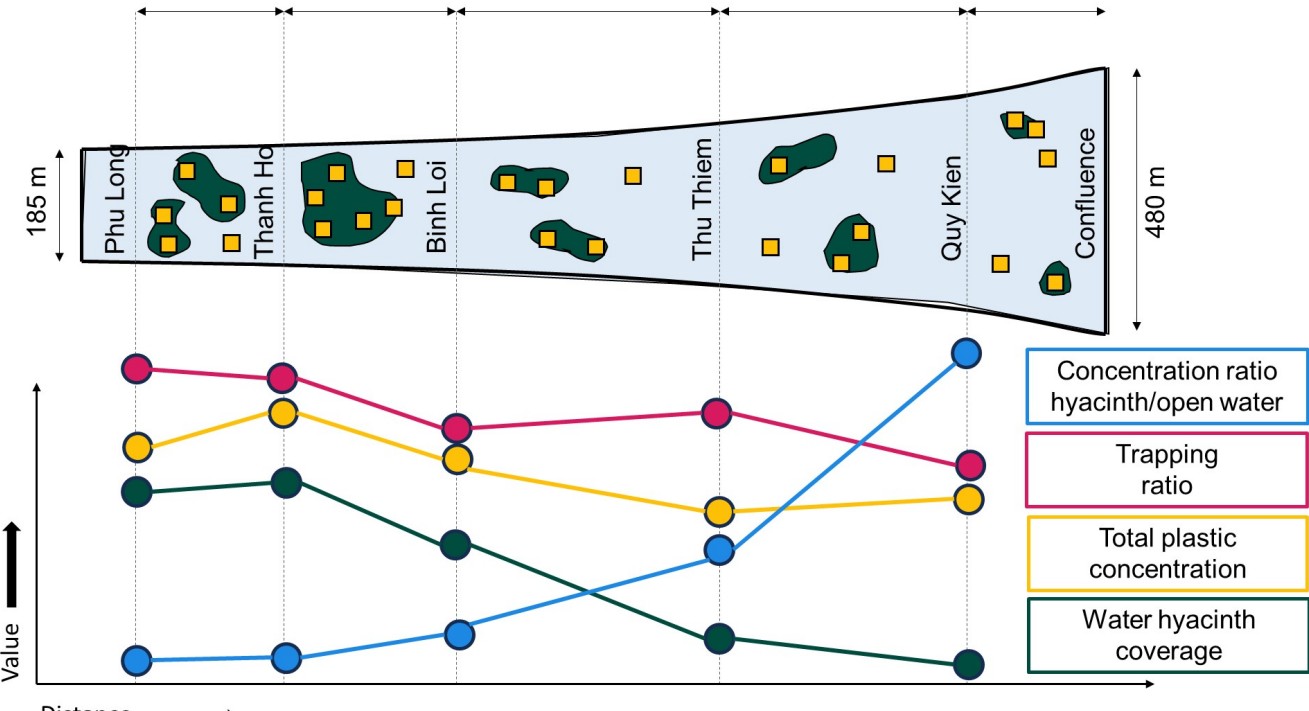

**Figure 3.** The plastic trapping is stable along the studied 42 km reach of the Saigon river. The water hyacinth coverage and total plastic surface concentration decreased towards the river mouth. The difference in surface concentration is largest in the most downstream location, suggesting the increased importance of water hyacinths in concentrating and carrying plastic pollution.

The application of the YOLOv8 deep learning model demonstrated both its strengths and limitations. YOLOv8 can only process images up to a resolution of 640x640 pixels. Consequently, different image processing methods are necessary based on the object dimensions of interest. In our case, the water hyacinths had much larger dimensions than the plastic objects. Literature recommends a DSG of 0.5 to 1.25 cm/pixel for macroplastic detection (Andriolo et al., 2023), and this range cannot be reached when the image resolution has to be reduced with a factor 32 (camera) to 56 (UAV). Our initial YOLOv8 configuration could not deal with this, and therefore we trained two separate models (i.e., Model$_{resize}$ and Model$_{tiles}$) using different image processing methods: (1) resizing images, and (2) cutting images into tiles. Developing a model with similar architecture for both plastic items and water hyacinths therefore remains a challenge.

We encourage exploring different deep learning algorithms and image processing strategies. For example, Wolf et al. (2020) developed an approach that classifies tiles within an image rather than bounding boxes for each object. They successfully distinguished between litter, vegetation, and other classes. With a $d_g$ of 0.2 cm/pixel, their images had a 2.3-12.3 times higher resolution than our image database. We therefore do not expect similarly high performance when applied to our dataset.





Another avenue for improvement is rethinking the classes that the model is trained on. We recommend to use only two classes, i.e. water hyacinth and plastic. This would make annotation less prone to error, as sometimes it is difficult to distinguish between free-floating and entrapped plastic items. As the model also would not have to distinguish between the two, it is expected that the overall performance would increase. The free-floating and entrapped plastic items could be identified in the post-processing phase, based on the overlap of the plastic and water hyacinth bounding boxes.

However, the low recall values suggest that we may underestimate the number of open water plastic items (factor 4), trapped plastic items (factor 2.5), and water hyacinths (factor 2). If we would correct our average statistics for this, the trapping ratio would decrease to 63%, and the water hyacinth coverage would double to 2.8%. The overall longitudinal patter would not change, although the trapping ratio would be lower at each location. The corrected trapping ratio would decrease from 81.9% to 73.9 % (Phu Long ), 82.0% to 74.0% (Thanh Ho), 63.9 to 52.5% (Binh Loi), 70.2% to 59.0% (Thu Thiem), and 58.1 to 46.4 % (Quy Kien). These values are still in line with previous work in the Saigon river, who found ranges in trapping ratio of 54-77% (Schreyers et al., 2024a) and water hyacinth coverage of 2-4% (Janssens et al., 2022).

## 3.5 Outlook

Water hyacinths have invaded rivers globally, mainly in (sub)tropical and even temperate zones (Téllez et al., 2008; VonBank et al., 2018; Lozano, 2021). The regions where water hyacinths are reported strongly overlap with the global distribution of the most polluted rivers (Meijer et al., 2021). We therefore expect that the findings in our paper are relevant for many other rivers around the world. Scientific evidence of similar plastic-water hyacinth interactions are still lacking, although there have been reports of similar findings including from the Citarum river, Indonesia (Pritasari Arumdati, 2021), and the Chao Phraya river, Thailand (Pajai, 2022). To better understand the transport and retention dynamics of river plastics, further work on plastic-water hyacinth interactions is required, also in other river systems globally.

It remains unresolved why water hyacinths are so effective in trapping plastics along rivers. Plastic may get stuck when individual water hyacinths form mats. From anecdotal evidence we also found large amounts of plastic towards the edges of the water hyacinths, suggesting the plastics mainly get stuck after the plants and plastics are in contact. Finally, plastics may also land on top of water hyacinths, especially for those which are mobilized by wind (Mellink et al., 2024). Equally, if not more, important is to better understand the potential release mechanisms. Plastic items may be released when water hyacinth patches disintegrate, or when the plastic-water hyacinth aggregates collide with riverbanks or vessels. The time and location of release are especially relevant for fate and transport modeling, and for optimizing plastic reduction interventions. We call for more fundamental work on describing and understanding the trapping and release dynamics of plastics in and from water hyacinths.

One of the crucial unknowns is how the surface area covered by water hyacinths and the plastic surface concentrations vary between and within rivers. In the Saigon river we found a rather constant trapping ratio for water hyacinth coverage between 0.1 and 4.0%. Globally, the surface area of river and lakes covered by water hyacinths has been found to vary greatly between rivers (10.6-98%) and lakes (30%) (Tobias et al., 2019; Moyo et al., 2013; Thamaga and Dube, 2019; Kleinschroth et al., 2021; Pádua et al., 2022). Whether water hyacinths still concentrate plastic as effectively at high surface coverage is still to



be assessed. In our study we looked at a limited river section of 42 km, and we recommend future work to further explore the distances over which plastic items can be carried by water hyacinths. Such insights are crucial to evaluate whether water hyacinths are mainly relevant around plastic source or retention areas, or along the full river extent.

The consistent plastic-water hyacinth interactions also has implication for river plastic pollution management. Despite the agreement that solutions should focus on upstream prevention, rather than plastic cleanup efforts (Bergmann et al., 2023), there are still ample global efforts to remove plastic pollution from rivers. Many of these initiatives use surface skimming vessels, or floating booms to concentrate and guide plastic towards an extraction point (Helinski et al., 2021). Consequently, plastic-water hyacinth aggregates may also be collected intentionally or unintentionally. As 63-82% of the total floating plastics is carried by water hyacinths, excluding those would result in missing a considerable share of the total plastics. However, collection should take into account the potential additional mass of the water hyacinths. As little as 0.5-3.2 % of the total sampled plastic-water hyacinth aggregate mass has been found the be plastic (van Emmerik et al., 2019; Schreyers et al., 2021b). The additional forces exerted by the water hyacinths may also result in damage to collection infrastructure (Cleanup, 2021). The high plastic concentrations in water hyacinths also offer opportunities. Water hyacinths are considered as a pest globally, and most control measures include mechanical and manual removal of plants (May et al., 2022; Karouach et al., 2022). Water hyacinth removal may therefore be used to also reduce floating plastic as a side effect.

In previous work we have coined the idea of using water hyacinths as proxy for river plastic (Schreyers et al., 2022). In recent years several methods have been developed to detect and monitoring water hyacinth invasion in rivers and lakes using optical (Janssens et al., 2022) and radar satellites (Simpson et al., 2022). By now there have been several examples of how water hyacinths can be monitored over large areas, for entire river systems lakes, and also using historical images. For example Mukarugwiro et al. (2021) and Kleinschroth et al. (2021) mapped water hyacinths across rivers and lakes using nearly 30 years of available imagery, showing relevant temporal changes in surface coverage. With reliable estimates of the water hyacinth plastic concentration and trapping ratio, the total abundance of floating plastics can be estimated (Schreyers et al., 2022). In our paper we show that the trapping ratio is rather constant along the river, but the water hyacinth plastic concentration varies around one order of magnitude. Field data on the trapping ratio and water hyacinth plastic concentration therefore remain crucial in further work on using water hyacinths as a proxy for plastic pollution. Note that such field calibration efforts are common practice in hydrology (stage-discharge relations) and sediment research (acoustic sensor calibration). The key challenge for the direct transfer of our findings to new rivers is building a reliable database of water hyacinth plastic concentration and trapping ratio values. Once more data is available, these values may also be estimated a priori by using statistical or conceptual models.

## 4 Conclusions

We quantified plastic concentration at the river surface and within water hyacinths at five locations along the 42 km most downstream section of the Saigon river, Vietnam, between February and April, 2023. A total of 15,495 images were collected using UAVs and cameras, and we used a YOLOv8 deep learning model to automatically detect plastic items and water hyacinths,





and quantify their numbers and measure the size of water hyacinth patches. We identified over 69k plastic items, and 57k water hyacinths.

Water hyacinths were found to be spatially consistent carriers of macroplastics. In total, 73% of the total detected plastic items were carried by water hyacinths, ranging between 63-82% for specific locations. The highest trapping ratio values were found most upstream, and most downstream of the studied river section. These results emphasize that water hyacinths play an

370 important role in the transport dynamics of plastic transport, both in time and space.

The studied section of the Saigon river had an average water hyacinth surface coverage of 1.4%, decreasing from 3.5% upstream to 0.2% in the most downstream section. Despite the low coverage compared to other rivers and lakes globally, the water hyacinths efficiently concentrate and aggregate floating macroplastics. The surface plastic density in water hyacinths was on average 109 times higher than at the open water surface, and increased from 24 to 306 in the downstream direction.

Our results emphasize the potential of using deep learning techniques to efficiently increase data availability . In turn, this allows to study complex environmental processes at larger spatial and temporal scales, for example for plastic-water hyacinth interactions. To date, previous research water plastic-water hyacinth interactions was mainly done using physical sampling, visual counting, or manual image processing methods. Our method combines images taken with off-the-shelf UAVs and cameras, and openly available software for image annotation and object detection, and can therefore easily be replicated

in other rivers globally.

With our paper we underscore the important role of water hyacinths in the transport and retention dynamics of macroplastic in rivers. Further work should focus on unravelling the capture and release dynamics of plastic items within water hyacinth plants and aggregates. Such insights are crucial to better understand the transport and fate of macroplastics in tropical rivers. Moreover, we encourage to further explore the potential of (1) using water hyacinths as proxy for plastic detection from space,

and (2) joint removal from rivers for plastic pollution reduction.

## Appendix A: Implementation of YOLOv8

We developed models with *Python 3.9.12* and *PyTorch 2.0.0* on a local NVIDIA GeForce RTX 3090 GPU (24GB). We firstly initialed the YOLOv8 with weights pre-trained on COCO dataset (Lin et al., 2014). Then, we fine-tuned the models on train subsets for 500 epochs and only saved the learning parameters yielding the highest valuation accuracy. We used the SGD

optimizer (Loshchilov and Hutter, 2016), with an initial rate of 0.01, an final value of 0.0001, a momentum of 0.937, weight decay of 0.001, and a batch size of 16. We set early stopping during training process when the validation accuracy does not improve in the last 50 epochs. We performed horizontal flipping augmentation method (Shorten and Khoshgoftaar, 2019) to improve model performance.



## Appendix B: Evaluation metrics

$$P = \frac{TP}{TP+FP} \tag{B1}$$

$$R = \frac{TP}{TP+FN} \tag{B2}$$

where $P$ is precision; $R$ is recall; $TP$ is true positive, representing the number of correctly detected positive cases; $FP$ is false positive, representing the number of cases incorrectly identified as positive; $FN$ is false negative, representing the number of ground-truth cases not detected by the model.

$$AP = \int_0^1 P(R)dR \tag{B3}$$

$$mAP = \frac{1}{n}\sum_{i=1}^{n} AP_i \tag{B4}$$

where $AP_i$ is the average precision of the $i$-th category, computed as the area under the precision-recall curve (Padilla et al., 2020); $n$ is the number of categories.

$$IoU = \frac{A_I}{A_U} \tag{B5}$$

where $IoU$ is Intersect over Union; $A_I$ and $A_U$ are the area of overlap and union between the ground-truth bounding box and the detected bounding box, respectively.



## Appendix C: Model detection performance

**Table C1.** Detection performance of (1) the $\text{Model}_{\text{resize}}$ on the $\text{Test}_{\text{resize}}$ subset, and (2) the $\text{Model}_{\text{tiles}}$ on the $\text{Test}_{\text{tiles}}$ subset

| Model | Class | Precision | Recall | (m)AP50 | (m)AP50-95[1] |
|---|---|---|---|---|---|
| $\text{Model}_{\text{resize}}$ | Water hyacinth | 0.83 | 0.54 | 70% | 48% |
| | Free-floating plastic | 0.72 | 0.05 | 39% | 24% |
| | Trapped plastic | 0.89 | 0.06 | 47% | 25% |
| | all classes | 0.81 | 0.21 | 52% | 32% |
| $\text{Model}_{\text{tiles}}$ | Water hyacinth | 0.74 | 0.45 | 60% | 44% |
| | Free-floating plastic | 0.72 | 0.25 | 48% | 37% |
| | Trapped plastic | 0.69 | 0.39 | 52% | 38% |
| | all classes | 0.72 | 0.37 | 53% | 40% |

[1] AP50-95 calculates the average AP at 10 different IoU thresholds varying in a range of 50% to 95%, with steps of 5%.







**Figure C1.** Detection results of the Model$_{resize}$ on the Test$_{resize}$ subset. The model can correctly detect many hyacinths, but it does not identify many plastic items in camera images ((a) and (b)) and drone images ((c) and (d)). Acronyms used: trapped plastic litter (ent-litter), Free-floating plastic litter (ff-litter).



## Appendix D: Plastic-water hyacinth interaction variables

The trapping ratio $r_{ent}$ [-] expresses the proportion of total plastic items trapped in water hyacinths. The total plastic items
equals the free-floating plastic items in open water $N_{free}$ [#], and the trapped plastic items in the water hyacinths $N_{wh}$ [#]:

$$r_{ent} = \frac{N_{ent}}{N_{ent} + N_{free}} \tag{D1}$$

The water hyacinth river surface coverage $f_{wh}$ [-] is the fraction of the total river surface $A_r$ [m$^2$] covered by water hyacinth
$A_{wh}$ [m$^2$]. For the total river surface area we used the total image FOV, which equals the sum of the open water surface and
water hyacinth surface area. The used the following equation:

$$f_{wh} = \frac{A_{wh}}{A_r} \tag{D2}$$

The total river plastic surface concentration $C_r$ [#/$m^2$] normalized the total number of plastic objects ($N_{ent} + N_{free}$) [#]
over the total river surface area $A_r$, which is equal to the image FOV [m$^2$]:

$$C_r = \frac{N_{ent} + N_{free}}{A_r} \tag{D3}$$

The water hyacinth plastic surface concentration $C_{wh}$ [#/$m^2$] is the total number of trapped plastic items $N_{ent}$ [#] normalized
over the total water hyacinth area per image $A_{wh}$ [m$^2$]:

$$C_{wh} = \frac{N_{ent}}{A_{wh}} \tag{D4}$$

The open water plastic surface concentration $C_o$ [#/$m^2$] is the total number of free-floating plastic items $N_{free}$ [#] normal-
ized over the total open water surface area (total FOV minus the water hyacinth area):

$$C_o = \frac{N_{free}}{(A_r - A_{wh}} \tag{D5}$$

Finally, to compare the differences between the plastic surface concentrations within water hyacinths, in the open water, and
in the river in total, we also calculate the ratios $C_{wf}/C_r$, and $C_{wf}/C_o$.



*Code and data availability.* All data will be made openly available through the 4.TU repository at https://tinyurl.com/WHY4tu. The code and the developed models for this study will be made available at https://github.com/TianlongJia/deep_plastic_YoloV8].

*Author contributions.* Conceptualization: TvE, TWJ, TJ, RT, LS; Methodology: TvE, TWJ, TJ, RT, LS; Software: TJ, RT, TWJ; Validation:
TvE, LS; Formal analysis: TWJ, TJ, LS; Investigation: TWJ, TJ, KB; Resources: TvE, KH, HN, RT; Data Curation: TvE, TWJ; Writing -
Original Draft: TvE, TWJ; Writing - Review & Editing: TWJ, TJ, KB, RT, HN, LS; Visualization: TWJ, LS, TvE, TJ; Supervision: TvE, LS,
RT; Project administration: TvE; Funding acquisition: TvE;

*Competing interests.* The authors declare that the research was conducted in the absence of any commercial or financial relationships that could be construed as a potential conflict of interest.

*Disclaimer.* N/A

*Acknowledgements.* The work of TvE was supported by the Veni Research Program, the River Plastic Monitoring Project with project number 18211, which was (partly) financed by the Dutch Research Council (NWO). The work of TJ was supported by China Scholarship Council (No. 202006160032) and the Directorate-General for Public Works and Water Management of The Netherlands (Rijkswaterstaat).



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
