# Peer review of "Water hyacinths as riverine plastic pollution carriers"

_EGUsphere, 2024_

## Referee Comment (RC2)

[referee-annotated manuscript omitted]

---

## Referee Comment (RC3)

Dear Authors,

The well-structured and written article introduces an innovative methodology for the detection of floating plastic in the Saigon river. It investigates the dynamic of water hyacinth and plastic at five sample sites along a 42 km section of the Saigon river in Vietnam using UAVs and bridge mounted fixed cameras. The applicability of the YOLOv8 deep learning model on tiled and resized imagery was evaluated. The plastic concentration was found to be higher in water hyacinth patches than in the open water. The water hyacinth surface cover decreased from the upstream to the downstream sample sites. However, the plastic concentration trapped within water hyacinth was the highest at the most downstream sampling site, which may suggest that the removal of plastic would be most efficient in the downstream region. Moreover, the article proves evidence for the importance of Water Hyacinth for the retention and transport of plastic at five different locations along the river.

Subsequently, you can find a list of comments that should be addressed prior to the publication.

**Minor comments:**

Line 67:          "using water hyacinths as a proxy"
As you state in line 226 & 227 considering the spatial and temporal variation of the water hyacinth trapped plastic concentration is important. In my opinion it is going to be challenging to use water hyacinth as a proxy for the quantification of riverine plastic due to the variability of different river systems and their drainage area leading to inconsistency in the amount or concentration of plastic trapped by water hyacinths.

Line 90:          Did you consider the flow speed of the Saigon and the resulting transport of water hyacinth and plastic to plan the field?
Were the field measurements planned and timed according to the flow speed to aim measuring the same patches of water hyacinth and plastic at all 5 sample sites?
If not the records may not be directly comparable due to the spatiotemporal variation of the water hyacinth and plastic concentration

Line 95:          How did you merge the records of the different measurement periods together?
                  Was imagery of the same measurement period used or did the measurement period vary for the different sample sites?

Figure 1.          "five measurement locations"
Why are you measuring at these locations?
Reasoning?
Hypothesis?
Why are you using bridge cameras at three location and UAV imagery at the other two?
Example:
"A bridge camera was installed at Binh Loi, since UAV flights are not permitted due to the no flight zone in the landing/departure zone of the nearby Tan Son Nhat airport."

Figure 1 A.          In the mini map presenting Vietnam and the neighboring countries, displaying the Saigon river and potentially Ho Chi Minh City would help readers that are not familiar with

the area to locate the study area.

On the map illustrating the Saigon River, displaying a rough outline of Ho Chi Minh city can create the link to the mini map in the upper right corner.

At the end of the article I was surprised to read that the studied river section traverses Ho Chi Minh City. Considering the map I was completely unaware of it. Knowing that Vietnam's largest city is right at the river banks may help readers to understand one potential source of the plastic pollution. For instance, the North arrow could be moved up north to the upper left corner, the scale down south to the lower left corner and the Legend over to the right below the mini map to make space to illustrate Ho Chi Minh city at the western bank of the Saigon.

Figure 1.        Caption and map's legend are not matching
no **crosses** (for UAV locations) in map's legend
**triangles** for UAV locations according to map, but for camera locations according to caption.

Figure 1.        In the text there is no cross reference to **Figure 1**. It would be nice to refer to the illustrations and map displayed in **Figure 1** in the relevant text passages. Examples:

**Fig. 1A** in **2.2 Field data collection**
for instance in line 91: "**Fig. 1A** illustrates the most upstream (Phu Long) and downstream (Quy Kien) locations, which were 41.9 km and 5.5 km upstream of [...]"

**Fig. 1B & Fig. 1D** in **2.2.1 Bridge mounted cameras**

**Fig. 1C & Fig. 1E in 2.2.2 Uncrewed Aerial Vehicle**
for instance in line 119: "UAV crossed the entire river width back and forth following a U-shape flight path (Geraeds et al., 2019) as illustrated in **Fig. 1C**."

Line 196:        Assumption of elliptical water hyacinth shape may not be applicable

Line 209:        Can the FOV corrected camera data from the bridges be directly compared to the UAV data? I there a potential limitation due to distortion/correction errors?

Figure 2.B.        "surface plastic concentration [#/km2] in water hyacinths increased towards the river mouth."        However, Quy Kien also has the biggest standard deviation or percentiles

Line 262:        "[...] traverses Ho Chi Minh City."
I expected much more plastic in the river after traversing Ho Chi Minh City. May this also partly explain the higher concentration of water hyacinth trapped plastic at Quy Kien?
May some of the plastic that was measured at previous sampling locations had been suspended/submerged or settled on the riverbed over the 42 km flow distance?
May the vertical transport of plastic have an effect within the observed 42 km of the river (van Emmerink & Schwarz 2020 Plastic debris in rivers) so that newly added plastic from Saigon is measured at Quy Kien and some of the plastic from upriver can't be observed since it is transported below the water surface?
In my opinion a short clarification about plastic transported by subsurface flows or deposited on the riverbed would be good to indicate that the application of your method may be limited to plastic floating close to the water surface but not the entire plastic within the river system. This clarification would also address the subsequent two points.

3.4 Uncertainties and limitations

Uncertainties about the amount of submerged/suspended plastic that can't be observed at the water surface?

**Schreyers et al. 2024** River plastic transport and storage budget

"suspended plastics account for over 96% of item transport within the river channel, while their relative contribution to mass transport is only 30%–37% (depending on the river section considered)."

Uncertainties about the time it takes for riverine or marine plastic to sink and not be detectable with the presented approach?

**van Emmerink & Schwarz 2020** Plastic debris in rivers

"foils and thin plastics, with a high surface area to mass ratio, tend to be affected more strongly by surface pollution, such as mud or biofouling, making the material heavier and more likely to sink or at least remain in the lower part of the water column"

"With low vertical transport, plastics remain more affected by horizontal transport. Lower in the water column, horizontal transport mechanisms are weaker and hence less pronounced"

Maybe the OEAN CLEANUP also published some research on the sinking rate of plastic

May a part of the plastic temporarily or permanently settle on the riverbed?

Figure C1.         "((a) and (b))

Influence of bridge's shadow on plastic and water hyacinth detection?

Based on RGB data do you think the shadow influence could interfere the detection compared to the sunny parts of the FOV?

**Technical comments to be addressed:**

Line 25:         Sub-Sahara**n**

Line 29:         "free-floating aquatic plants with freely hanging roots"
                **Venter et al. 2017:**
                **"[…] it occurs as free-floating plant or to a lesser extent as an emergent macrophyte (Penfound and Earle, 1948, Barret and Forno, 1982)."**
                "Water hyacinth (Eichhornia crassipes Mart. Solms Pontederiaceae) **mainly occurs as a free-floating aquatic plant**, but **can survive decreasing water levels when rooted in soil**. This adaptation to seasonal fluctuations in hydrology may contribute to its invasive potential in natural and man-made water bodies, where **stranded plants can take root**."

Line 39:         van Emmerik et al., **(**2019)

Line 99:         (Table 2)

Line 99:         pointing

Line 162:        delete one being from being being

Line 185:     "The results show [...]"
The Model performance evaluation results are already presented in the Methods section. Maybe it would be better to present the results in the results section.

Line 217:     delete 'at'

Figure 2.     add 'D. '        [...] towards the river mouth. D. The ratio between [...]

Line: 222     write    at     instead of and

Line 255:     add 'be'        may 'be' caused

Line 259:     either delete parentheses or delete '(variations)'

Line 307:     add 'n'         patter**n**

Line 351:     add a comma ','       "river systems',' lakes, and [...]"

Line 390:     delete 'n'              "a final value of [...]"

Line 414:     replace 'The' with 'We'        "We used the following equation:"

Line 424:     add closing parantheses ')'     (A_r - A_wh')'

**Access review (quick report), peer review, and interactive public discussion**

1.  Does the paper address relevant scientific questions within the scope of BG?
    Yes.

2.  Does the paper present novel concepts, ideas, tools, or data?
    Yes, data on free floating and trapped plastic at 5 sampling sites along the Saigon river.

3.  Are substantial conclusions reached?
    Yes.

4.  Are the scientific methods and assumptions valid and clearly outlined?
    Yes.

5.  Are the results sufficient to support the interpretations and conclusions?
    Yes.

6.  Is the description of experiments and calculations sufficiently complete and precise to allow their reproduction by fellow scientists (traceability of results)?
    Yes.

7.  Do the authors give proper credit to related work and clearly indicate their own new/original contribution?
    Yes, previous work by for instance Schreyer et al. & van Emmerink was mentioned and clarified.

8.  Does the title clearly reflect the contents of the paper?
    Yes.

9. Does the abstract provide a concise and complete summary?
   Yes.

10. Is the overall presentation well structured and clear?
    Yes.

11. Is the language fluent and precise?
    Yes.

12. Are mathematical formulae, symbols, abbreviations, and units correctly defined and used?
    Yes.

13. Should any parts of the paper (text, formulae, figures, tables) be clarified, reduced, combined, or eliminated?
    Yes, consider the comments.

14. Are the number and quality of references appropriate?
    Yes.

15. Is the amount and quality of supplementary material appropriate?
    Yes.

This review was written by an early career scientist.

---

## Author Comment (AC1)

**Reply to RC1**

Tim Hans Martin van Emmerik, Tim Willem Janssen, Tianlong Jia, Thank-Khiet L. Bui, Riccardo Taormina, Hong-Q. Nguyen, and Louise Jeanne Schreyers

**Reviewer comments**

The article is well-written; no major issues were found during my review. Below is a list of a few minor issues and technical corrections the authors should address before publication.

**Thank you for your positive and constructive feedback. Please find our response in bold.**

1. Page 4, caption of Fig. 1: there is a mismatch between what is said in the caption and what is reported in Panel A, i.e.: "The triangles indicate the camera locations, and the crosses the UAV locations", on the map instead I think that circles indicate "camera locations" while triangles indicate "UAV locations", right? Please correct.

**We will correct this.**

2.      Line 162: the word "being" is repeated twice. Please remove.

**We will correct this.**

3.      Line 204: this should be "Kruskal–Wallis" right?

**Correct, this should be Kruskall-Wallis.**

4.      Line 259: …or that water hyacinths could get stuck and trapped on riverbanks upstream and stop flowing downstream? Is this another option that would explain this observation right? In other words, a large fraction is trapped before and this would explain why fewer plants are detected downstream right? However in this case, also the plastic would be trapped in river banks right? Can the authors briefly discuss and elaborate on these points?

**Thanks for the suggestion. We will include this in the revised manuscript.**

5.      Line 306: "pattern"

**We will correct this.**

---

## Author Comment (AC2)

**Reply to RC2**

Tim Hans Martin van Emmerik, Tim Willem Janssen, Tianlong Jia, Thank-Khiet L. Bui, Riccardo Taormina, Hong-Q. Nguyen, and Louise Jeanne Schreyers

**Reviewer comments**

Please check the annotated manuscript for revisions that are mainly about forms and light clarifications. The manuscript is well written and clear.

**Thank you for your positive and constructive feedback. Please find our reply in bold.**

Specific comments

1. suggestion: debris

**We will use aligned terminology in the revised manuscript.**

2. surface plastic concentration? please specify

**We will correct this.**

3. what was the smallest recognizable item detected by the camera ?

**Assuming that 8-10 pixels are required to detect an item, the minimum detectable item size was 2.2-2.8 cm for the lowest ground sampling distance of 0.28 cm/pixel. We will include this.**

4. at what place/station?

**We will add the minimum detectable item size per location to Table 2.**

5. pleade specify now the duration. Should be 310 seconds ?

**We will include this. This was indeed 310 seconds, plus some time in between (7-8 minutes per measurement).**

6. Could you explain the interest of this measurement in one sentence please ?

**The Ground Sampling Distance is relevant as it estimates the size of each pixel, which determines the theoretical minimum detectable item size. We aimed to decrease the variation between $d\_g$ between locations to increase the comparability of the datasets. We will add this to the revised manuscript.**

7. Could you specify in the caption what is annotated images and items and to what purpose ? In case the reader miss the beginning where it is writtent "training algorithm YOLOv8

**Thanks for the suggestion, we will add this.**

8. I count 9365 in Table 2 Please check the amount reported.

**This refers to annotated objects in Table 3, we will clarify this in the text.**

9. is it related to what you explain in section 2.3.3. ? If yes, please indicate it, because it is the first occurrence of "resize" and we do not really understand why before 2.3.3.

**We will clarify this.**

10. Pkease specify an order of magnitude. What is "low"? 10 cm, 1 cm ?

**We will clarify this.**

11. Ok, very clear!

**We will rearrange so that Table 3 comes after this.**

12. Expected regarding what you clearly explained in section 2.3.3. But it is not clear if you decided to use model resize to account for water hyacinths only, and model tiles to account for plastic debris only ?

**The Model-resize is only for the water hyacinths, and the Model-tiles for the plastic items only. We will clarify this in the revised manuscript.**

13. you mean Wallis ?

**Yes, we will clarify this.**

14. please specify again in the caption what is Cwh, C0 and Cr

**We will clarify this.**

15. Schreyers et al. evaluated Cwh in similar river systems. So, I would be more cautious using Cwh 2.1 .10^5 #/km2 for other tropical systems.

**We will add that the estimates of Schreyers et al. (2022) for the same river are based on visual counting measurements, and not images. We will clarify this.**

16. But is it a problem for accuracy of detection ? Is there any differences in size of plastic debris detected depending on the location \(i.e., height, cam parameters, etc.\)?

**Not for the accuracy of detection, but this may introduce uncertainty in describing and understanding the system as a whole using the (equally) accurate data across locations. We will clarify this.**

17. yes. Here it would be interesting to put some quantifiable sizes. Because even a small difference in size detection might introduce large quantification biases if key size classes are concerned like mesoplastics that might represent a ve\ rgy high amount in number but also in mass.

**We will add these to Table 2 and refer to the values in this section.**

18. I see a kind of contradiction in terms when writting : "3. The plastic trapping is stable \(...and...\) increased importance of wate\ r hyacinths in concentrating and carrying plastic pollution

[downstream]".  If the concentration remain nearly stable, so the importance of W\ H would decrease with their decreasing coverage. Perhaps you should switch from relative to absolute quantification here ?

**We try to make the point that the role of water hyacinths becomes more important in the downstream areas, as their relative concentration increases. The absolute concentration remains stable indeed, so we will clarify this in the revised manuscript.**

19.      missing something between water and plastic ?

**"on" is missing, we will correct this.**

20.      Hmm those are very low... This might be the strongest limitation

**We will further elaborate on this in the revised discussion.**

21.      In text you refer to km2 Please make sure all conversions are good or harmonize

**Here we use the river and water hyacinth surface to calculate a dimensionless number, so either m2 or km2 works. For consistency we will use km2 here too.**

22.      Not mandatory suggestion Draw a schematic river \(for exemple like in fig 3\) to show the different variables in context

**Thanks for the suggestion. We will consider including a figure as suggested.**

---

## Author Comment (AC3)

**Reply to RC3**

Tim Hans Martin van Emmerik, Tim Willem Janssen, Tianlong Jia, Thank-Khiet L. Bui, Riccardo Taormina, Hong-Q. Nguyen, and Louise Jeanne Schreyers

**Reviewer comments**

Dear Authors, The well-structured and written article introduces an innovative methodology for the detection of floating plastic in the Saigon river. It investigates the dynamic of water hyacinth and plastic at five sample sites along a 42 km section of the Saigon river in Vietnam using UAVs and bridge mounted fixed cameras. The applicability of the YOLOv8 deep learning model on tiled and resized imagery was evaluated. The plastic concentration was found to be higher in water hyacinth patches than in the open water. The water hyacinth surface cover decreased from the upstream to the downstream sample sites. However, the plastic concentration trapped within water hyacinth was the highest at the most downstream sampling site, which may suggest that the removal of plastic would be most efficient in the downstream region. Moreover, the article provides evidence for the importance of Water Hyacinth for the retention and transport of plastic at five different locations along the river.

**Thank you for your positive, detailed, and constructive feedback. Please find our reply in bold.**

Subsequently, you can find a list of comments that should be addressed prior to the publication.

Minor comments:
1. Line 67: "using water hyacinths as a proxy" As you state in line 226 & 227 considering the spatial and temporal variation of the water hyacinth trapped plastic concentration is important. In my opinion it is going to be challenging to use water hyacinth as a proxy for the quantification of riverine plastic due to the variability of different river systems and their drainage area leading to inconsistency in the amount or concentration of plastic trapped by water hyacinths.

**To date, studies on water hyacinth-plastic interactions have been limited to the Saigon river. However, there is additional anecdotal evidence from other rivers in Thailand, Dominican Republic, Vietnam, and Indonesia that water hyacinths trap considerable proportions of the total plastic pollution. We agree that it will be challenging to infer plastic concentrations from water hyacinth observations without any further information or field data. Our long-term goal is to explore how water hyacinths and plastic concentration are correlated across river systems. This could be used to derive river-scale or site-specific empirical relationships between water hyacinths and plastics, similar to rating curves (relationship between water level and discharge) or sediment load equations. In addition, we use the co-occurrence of water hyacinths and plastic to explore whether water hyacinths may be a good indicator of where plastic is located within a river system. We will include this perspective in the revised manuscript.**

2. Line 90: Did you consider the flow speed of the Saigon and the resulting transport of water hyacinth and plastic to plan the field? Were the field measurements planned and timed according to the flow speed to aim measuring the same patches of water hyacinth and plastic at all 5 sample sites? If not the records may not be directly comparable due to the spatiotemporal variation of the water hyacinth and plastic concentration

**Flow velocity was not measured during the fieldwork due to practical constraints. The Saigon has a strong tidal influence, with flow reversal twice a day. To make sure the data collected at different locations and times are comparable, we covered both ebb and flood tide equally. On each day, we took measurements for 3-4 hours in the morning and 3-4 in the afternoon, covering close to a full tidal cycle. We did not aim to measure the exact same patches at each sampling site, and rather focused on establishing robust statistics on plastic, water hyacinths, and their interactions for each site and observation round.**

3.     Line 95: How did you merge the records of the different measurement periods together? Was imagery of the same measurement period used or did the measurement period vary for the different sample sites?

**We calculated the average values using all observations per site. The error bar shows the min/max value during the full observation period. We will include more details on the data processing steps, and clarify this in the text and caption.**

4.     Figure 1. "five measurement locations" Why are you measuring at these locations? Reasoning? Hypothesis? Why are you using bridge cameras at three location and UAV imagery at the other two? Example: "A bridge camera was installed at Binh Loi, since UAV flights are not permitted due to the no flight zone in the landing/departure zone of the nearby Tan Son Nhat airport."

**We used UAVs to increase the spatial coverage of the data collection. In between Phu Long and Binh Loi, and downstream of Thu Thiem, no suitable bridges were available for camera-based measurements. Thanh and Quy Kien were selected as these were the most suitable locations for which permission was granted. No additional UAV location was available between Binh Loi and Thu Thiem. We will add a brief rationale in the revised manuscript.**

5.     Figure 1 A. In the mini map presenting Vietnam and the neighboring countries, displaying the Saigon river and potentially Ho Chi Minh City would help readers that are not familiar with the area to locate the study area. On the map illustrating the Saigon River, displaying a rough outline of Ho Chi Minh city can create the link to the mini map in the upper right corner. At the end of the article I was surprised to read that the studied river section traverses Ho Chi Minh City. Considering the map I was completely unaware of it. Knowing that Vietnam's largest city is right at the river banks may help readers to understand one potential source of the plastic pollution. For instance, the North arrow could be moved up north to the upper left corner, the scale down south to the lower left corner and the Legend over to the right below the mini map to make space to illustrate Ho Chi Minh city at the western bank of the Saigon.

**Thanks for the suggestions, we will update the map in the revised manuscript.**

6.     Figure 1. Caption and map's legend are not matching no crosses (for UAV locations) in map's legend triangles for UAV locations according to map, but for camera locations according to caption.

**We will revise the caption and legend accordingly.**

7.     Figure 1. In the text there is no cross reference to Figure 1. It would be nice to refer to the illustrations and map displayed in Figure 1 in the relevant text passages. Examples: Fig. 1A in 2.2 Field data collection for instance in line 91: "Fig. 1A illustrates the most upstream (Phu Long) and downstream (Quy Kien) locations, which were 41.9 km and 5.5 km upstream of [...]"

**We will correct this.**

8.     Fig. 1B & Fig. 1D in 2.2.1 Bridge mounted cameras Fig. 1C & Fig. 1E in 2.2.2 Uncrewed Aerial Vehicle for instance in line 119: "UAV crossed the entire river width back and forth following a U-shape flight path (Geraeds et al., 2019) as illustrated in Fig. 1C."

**We will correct this.**

9.     Line 196: Assumption of elliptical water hyacinth shape may not be applicable

**We agree, and will include this in the methods and discussion of the revised manuscript.**

10.     Line 209: Can the FOV corrected camera data from the bridges be directly compared to the UAV data? I there a potential limitation due to distortion/correction errors?

**Comparing images with varying angles, ground sampling distance, and FOV, may lead to uncertainty. However, we found that the min/max ground sampling distance is rather stable over time and space (Table 2). There are some outliers, mainly due to the initial high angle of the camera at Thu Thiem. We therefore deem the data good enough for comparison, but we will expand on this in the discussion.**

11.     Figure 2.B. "surface plastic concentration [#/km2] in water hyacinths increased towards the river mouth." However, Quy Kien also has the biggest standard deviation or percentiles

**Correct, we will add that to the revised manuscript.**

12.     Line 262: "[…] traverses Ho Chi Minh City." I expected much more plastic in the river after traversing Ho Chi Minh City. May this also partly explain the higher concentration of water hyacinth trapped plastic at Quy Kien? May some of the plastic that was measured at previous sampling locations had been suspended/submerged or settled on the riverbed over the 42 km flow distance? May the vertical transport of plastic have an effect within the observed 42 km of the river (van Emmerink & Schwarz 2020 Plastic debris in rivers) so that newly added plastic from Saigon is measured at Quy Kien and some of the plastic from upriver can't be observed since it is transported below the water surface? In my opinion a short clarification about plastic transported by subsurface flows or deposited on the riverbed would be good to indicate that the application of your method may be limited to plastic floating close to the water surface but not the entire plastic within the river system. This clarification would also address the subsequent two points.

**These are relevant points and we will add a clarification to the section.**

13.     3.4 Uncertainties and limitations
        Uncertainties about the amount of submerged/suspended plastic that can't be observed at the water surface?
        Schreyers et al. 2024 River plastic transport and storage budget "suspended plastics account for over 96% of item transport within the river channel, while their relative contribution to mass transport is only 30%–37% (depending on the river section considered)."
        Uncertainties about the time it takes for riverine or marine plastic to sink and not be detectable with the presented approach?
        van Emmerink & Schwarz 2020 Plastic debris in rivers "foils and thin plastics, with a high surface area to mass ratio, tend to be affected more strongly by surface pollution, such as mud or biofouling, making the material heavier and more likely to sink or at least remain in the lower part of the water column" "With low vertical transport, plastics remain more affected by horizontal transport. Lower in the water column, horizontal transport mechanisms are weaker and hence less pronounced" Maybe the OEAN CLEANUP also published some research on the sinking rate of plastic May a part of the plastic temporarily or permanently settle on the riverbed?

**We will add a paragraph on the uncertainties due to missing suspended plastics with our observation methods.**

14.     Figure C1. "((a) and (b)) Influence of bridge's shadow on plastic and water hyacinth detection? Based on RGB data do you think the shadow influence could interfere the detection compared to the sunny parts of the FOV?

**Shadow can play a role in object detection, but generally has a lower impact compared to sunglint, foam, or ripples.**

Technical comments to be addressed:
    1.   Line 25: Sub-Saharan

**We will correct this.**

2.       Line 29: "free-floating aquatic plants with freely hanging roots" Venter et al. 2017: "[...] it occurs as free-floating plant or to a lesser extent as an emergent macrophyte (Penfound and Earle, 1948, Barret and Forno, 1982)." "Water hyacinth (Eichhornia crassipes Mart. Solms Pontederiaceae) mainly occurs as a free-floating aquatic plant, but can survive decreasing water levels when rooted in soil. This adaptation to seasonal fluctuations in hydrology may contribute to its invasive potential in natural and man-made water bodies, where stranded plants can take root."

**We will adapt the sentence.**

3.       Line 39: van Emmerik et al., (2019)

**We will correct this.**

4.       Line 99: (Table 2)

**We will correct this.**

5.       Line 99: pointing

**We will correct this.**

6.       Line 162: delete one being from being being

**We will correct this.**

7.       Line 185: "The results show […]" The Model performance evaluation results are already presented in the Methods section. Maybe it would be better to present the results in the results section.

**We will reconsider where to present the model performance evaluation.**

8.       Line 217: delete 'at'

**We will correct this.**

9.       Figure 2. add 'D. ' […] towards the river mouth. D. The ratio between [...]

**We will correct this.**

10.      Line: 222 write at instead of and

**We will correct this.**

11. Line 255: add 'be' may 'be' caused

**We will correct this.**

12. Line 259: either delete parentheses or delete '(variations)'

**We will correct this.**

13. Line 307: add 'n' pattern

**We will correct this.**

14. Line 351: add a comma ',' "river systems',' lakes, and […]"

**We will correct this.**

15. Line 390: delete 'n' "a final value of […]"

**We will correct this.**

16. Line 414: replace 'The' with 'We' "We used the following equation:"

**We will correct this.**

17. Line 424: add closing parantheses ')' (A_r - A_wh')'

**We will correct this.**